# A Comprehensive Approach Combining Short-Chain Polyphosphate and Bacterial Biostimulants for Effective Nutrient Solubilization and Enhanced Wheat Growth

**DOI:** 10.3390/microorganisms12071423

**Published:** 2024-07-13

**Authors:** Kaoutar Bourak, Fatima Ezzahra Oulkhir, Fatima Zahra Maghnia, Sebastien Massart, Latefa Biskri, M. Haissam Jijakli, Abdelmounaaim Allaoui

**Affiliations:** 1Microbiology Laboratory, Mohammed VI Polytechnic University, Lot 660, Hay Moulay Rachid, Benguerir 43150, Morocco; bourakkaoutar@gmail.com (K.B.);; 2Integrated and Urban Plant Pathology Laboratory, Terra Research Center, Gembloux Agro-Bio-Tech, Liege University, 5030 Gembloux, Belgium; andalous_fz@hotmail.com (F.Z.M.); sebastien.massart@uliege.be (S.M.)

**Keywords:** rhizosphere, *Bacillus tropicus*, Pseudomonas, polyphosphate fertilizer, P solubilization, K solubilization, Zn solubilization, biocontrol activity, wheat growth

## Abstract

Phosphorus constitutes a crucial macronutrient for crop growth, yet its availability often limits food production. Efficient phosphorus management is crucial for enhancing crop yields and ensuring food security. This study aimed to enhance the efficiency of a short-chain polyphosphate (PolyP) fertilizer by integrating it with plant growth-promoting bacteria (PGPB) to improve nutrient solubilization and wheat growth. Specifically, the study investigated the effects of various bacterial strains on wheat germination and growth when used in conjunction with PolyP. To achieve this, a greenhouse experiment was conducted in which the wheat rhizosphere was amended with a short-chain PolyP fertilizer. Based on the morphological aspect, eight bacteria, designated P1 to P8, were isolated and further characterized. Plant growth-promoting traits were observed in all bacterial strains, as they presented the ability to produce Indole Acetic Acid (IAA) in significant amounts ranging from 7.5 ± 0.3 µg/mL to 44.1 ± 2 µg/mL, expressed by *B. tropicus* P4 and *P. soyae* P1, respectively. They also produced ammonia, hydrogen cyanide (HCN), and siderophores. Their effect against the plant pathogen *Fusarium culmorum* was also assessed, with *P. reinekei* P2 demonstrating the highest biocontrol activity as it presented a total inhibitory effect. Additionally, some strains exhibited the ability to solubilize/hydrolyze phosphorus, potassium, and zinc. In vivo, the initial growth potential of wheat seeds indicated that those inoculated with the isolated strains exhibited elevated germination rates and enhanced root growth. Based on their plant growth-promoting traits and performance in the germination assay, three strains were selected for producing the best results, specifically phosphorus hydrolyzation/solubilization, zinc solubilization, IAA production, HCN, and siderophores production. Wheat seeds were inoculated by drenching in a bacterial suspension containing 10^10^ CFU/mL of log phase culture, and an in planta bioassay was conducted in a growth chamber using three selected strains (*Pseudomonas soyae* P1, *Pseudomonas reinekei* P2, and *Bacillus tropicus* P4), applied either individually or with PolyP on a P-deficient soil (28 mg/kg of P Olsen). Our findings demonstrated that the combination of *Pseudomonas soyae* P1 and PolyP achieved the highest shoot biomass, averaging 41.99 ± 0.87 g. Notably, applying *P. soyae* P1 or *Bacillus tropicus* P4 alone yielded similar results to the use of PolyP alone. At the heading growth stage, the combination of *Bacillus tropicus* P4 and PolyP significantly increased the Chlorophyll Content Index (CCI) to 37.02 µmol/m^2^, outperforming both PolyP alone (24.07 µmol/m^2^) and the control (23.06 µmol/m^2^). This study presents an innovative approach combining short-chain PolyP with bacterial biostimulants to enhance nutrient availability and plant growth. By identifying and characterizing effective bacterial strains, it offers a sustainable alternative to conventional fertilizers.

## 1. Introduction

Phosphorus (P) plays a significant role in plant growth and development through many biological processes, including energy transfer, cell division, and the modulation of gene expression [1,2]. It is often necessary to apply P fertilizers to provide plants with the nutrients they need to grow and thrive. Several P-based fertilizers, including polyphosphate fertilizer (PolyP), have proven their capacity to improve plant growth and crop yields [3,4]. PolyP fertilizers are small releases of anionic polymers composed of phosphoanhydride bonds connecting two to hundreds of orthophosphate residues (OrthoP) creating either linear chains (linear PolyP), cyclic chains (meta PolyP), or branching structures (ultra PolyP) [5,6]. PolyP fertilizer releases P over a longer period of time than orthophosphate fertilizer. They are designed to deliver P to plants more efficiently [7]. 

Given the critical role that phosphorus plays in crop health, any limitations in its availability can significantly impact food production. This is particularly evident in crops like wheat, which is a staple food in human diets and a crucial component of livestock feed for meat production [8]. Thus, any reduction in wheat production would have an immediate effect on the food supply. The bioavailability of P from PolyP in the soil can vary significantly due to their inherent properties (such as their chemical structure, physical form, and chain length) and the characteristics of the soil (such as temperature, pH, texture, moisture content, and microbial activity) [3,9,10]. P concentrations in soil can vary widely, ranging from 100 to 3000 mg/kg, mostly as mineral complexes corresponding to tricalcium phosphate (Ca_3_ (PO_4_)_2_), iron phosphate (FePO_4_), aluminum phosphate (AlPO_4_) [11], and as phytate, which accounts for the vast majority of soil organic P [12]. Therefore, the portion of phosphorus readily available for plant growth exists in two forms: H_2_PO_4_^-^ (dihydrogen phosphate) and HPO_4_^2-^ (monohydrogen phosphate), known as the “bio-available P fraction”, which is often much lower, typically in the 0.1 to 10 mg/kg range. This bio-available fraction does not support plant growth [13,14,15]. 

Phosphate-solubilizing bacteria (PBS) belong to the plant growth-promoting bacteria (PGPB) group and are microorganisms living in the rhizosphere that make P available for plants. They have attracted significant attention recently due to their potential to improve plant growth and development. They can affect the availability of phosphorus through the microbial decomposition of organic matter, releasing phosphorus from organic sources, making it available for plant uptake, or producing enzymes that help plants take up phosphorus more efficiently [16,17]. They used three main P solubilization mechanisms: (1) the release of complexing or mineral dissolving compounds, such as organic acid anions, siderophores, protons, hydroxyl ions, and CO_2_. (2) Extracellular enzyme liberation (biochemical P mineralization); microorganisms secrete enzymes called phytases into the soil. These phytases catalyze the breakdown of phytates, which are the major organic phosphorus compounds in soil. By liberating inorganic phosphates from phytates through enzymatic action, microorganisms can compensate for plants’ inability to directly acquire phosphorus from phytates (Richardson and Simpson 2011) [18]. (3) P release during substrate degradation (biological P mineralization) [19]. As a result, microbes play a critical role in all three primary components of the soil P cycle (dissolution-precipitation, sorption-desorption, and mineralization-immobilization).

PGPB has the potential to improve plant growth and development through several mechanisms. First, they can directly promote nutrient acquisition. In addition to P solubilization, some bacteria, such as potassium solubilizing bacteria (KSB) and zinc solubilizing bacteria (ZSB), can solubilize potassium (K) and zinc (Zn), respectively. Second, a range of substances and mechanisms that support plant development can be produced by PGPB. These include: (1) phytohormones controlling a variety of factors of plant development and growth [20] (e.g., Indole acetic acid (IAA), when sprayed on wheat seedlings, has been shown to improve germination, radicle and coleoptile length, leaf relative water content, biochemical responses, and physiological and molecular responses [21]); and (2) nitrogen fixation (the process by which molecular nitrogen (N_2_), with a strong triple covalent bond, is converted into ammonia (NH_3_) or related nitrogenous compounds), which is essential for the biosynthesis of all nitrogen-containing organic compounds, including amino acids, proteins, and nucleic acids; crucial for agriculture and the manufacture of fertilizer [22,23,24,25]. Some PGPB are endowed with biocontrol activity against phytopathogens, which is considered an effective method to treat plant diseases in an eco-friendly manner without causing harm to the environment. They may produce hydrogen cyanide (HCN), which acts as a metabolic inhibitor against pathogens and fungal activity, thereby enhancing their biocontrol effectiveness. Additionally, these microorganisms can produce siderophores, which are iron-chelating compounds with various applications, including promoting plant growth and controlling plant pathogens [24,25]. These small molecules play a crucial role in iron acquisition and can have significant impacts on agriculture. Therefore, there is interest in characterizing other beneficial properties of P-solubilizing bacteria to harness their potential as beneficial microorganisms.

In a previous study, we revealed that PolyP application positively affects wheat physiological traits directly related to yield and significantly influences bacterial and fungal microbiota in the rhizosphere and rhizoplane at different growth stages compared to orthophosphate fertilizer [26]. In this study, we proposed two hypotheses: First, adding PolyP fertilizer to the wheat rhizosphere would encourage the growth of bacterial strains that enhance plant growth. Second, the selected plant growth-promoting bacteria (PGPB) would have a synergistic effect on wheat growth when used together with PolyP.

## 2. Material and Methods

### 2.1. Soil Sampling and Preparation 

Soil samples were taken from the top layer of the soil (0–20 cm) of an agricultural site (32°13′12.2″ N 7°53′34.9″ W) located in Benguerir, Morocco. The sampling site was chosen based on its low P soil content to support wheat growth (28 mg/kg of P Olsen) [27]. Soil, texture, and chemical analyses were performed. The studied soil corresponded to a sandy clay with 435 mg/kg of K_2_O, 0.09 % of total N (2.49 mg/kg of N-NH_4_ and 61.53 mg/kg of N-NO_3_), 0.63 mg/kg of Cu, a pH of water 8.41, 0.27 mS/cm of electric conductivity at 25 °C, 0.63 mg/kg of Zn, 388 mg/kg of MgO, 3.18 mg/kg of Fe, 116 mg/kg of Na, and 1.46% of organic matter. The soil was sieved at 8 mm and homogenized with sand (4:1, *v*/*v*). The experimentation was carried out in the Agriculture Innovation and Technology Transfer Center greenhouse of Mohammed VI Polytechnic University (UM6P), Morocco. 

### 2.2. Applied PolyP Fertilizer

PolyP fertilizer is a linear, short-chain compound composed entirely of tripolyphosphate and has a P_2_O_5_ content of 47%. 

### 2.3. Isolation of Rhizosphere Bacterial Strains

The express purpose of this experiment was to isolate PSB strains. A Moroccan bread wheat (Faiza variety, *Triticum aestivum* L.) was used. The amount of fertilizer applied was decided upon by considering the findings of the soil analysis as well as the nutritional needs of bread wheat, which include nitrogen (N), phosphorus (P), and potassium (K) at rates of 180, 60, and 150 mg/kg of dry soil, respectively. Samples of wheat roots were taken at development stage Z69 from each pot (Zadoks et al., 1974) [28]. Before being used, each sample was kept at −20 °C in a sterile bag. To foster the growth of PSB strains, five seeds were planted in each pot and modified with PolyP in ten repetitions.

Two grams (g) of roots, pooled from 10 repetitions, were washed using Kalium Phosphate Buffer and Tween (KPBT) plus ultra-bath sonication (Mod. AU-32, Argo laboratory, Carpi MO, Italy) for 5 min [29]. Then, serial dilutions (10^−1^ to 10^−5^) using MilliQ water (Milli-Q^®^, Merck KGaA, Darmstadt, Germany) were generated and further spread on a Trypticase Soy Agar (TSA) (BIOKAR Diagnostics, Beauvais, France) medium. Plates were then incubated at 28 °C for 24 h. Emerging colonies were subcultured to obtain pure isolates that were stored in cryotubes at −80 °C while being cryoprotected with 10% dimethyl sulfoxide (DMSO) (Merck KGaA, Darmstadt, Germany). 

### 2.4. Species Determination of the Isolated Strains 

The bacterial DNA of each strain was extracted from overnight culture in Luria-Bertani (L.B) broth (Thermo Fisher Scientific, Osterode Am Harz, Germany) using an Invitrogen PureLink Genomic DNA Mini Kit (Thermo Fisher Scientific, Osterode Am Harz, Germany) according to the manufacturer’s instructions.

Direct PCR was made using a Bioline PCR kit (Bioline, Luckenwalde, Germany) using 16SA1 (5-AGAGTTTGATCMTGGCTCAG-3) and 16SB1 (5-TACGGYTACCTTGTTACGACTT-3) primers, allowing the amplification of the 16S rDNA [30]. PCR was performed in a 25 µl reaction containing 5 µL of buffer (5×), 0.75 µL of MgCl_2_ (50 mM), 0.5 µL of dNTP mix (10 mM), 0.5 µL of each primer (10 mM), 0.5 µL of Mango Taq polymerase (1000 U/µL), 16.25 µL of water, and 1 µL of DNA template. Then, 35 amplification cycles began with an initial denaturation phase at 94 °C for 5 minutes (min). This was followed by 35 cycles, each comprising denaturation at 94 °C for 30 seconds (s), annealing at 54 °C for 30 s, and elongation at 72 °C for 1 min. The final step involved an extension phase set at 72 °C for 10 min [30]. Amplicons were then purified using a QIAquik PCR Purification Kit (QIAGEN, Hilden, Germany) and sent to Macrogen Europe (Amsterdam, The Netherlands) for Sanger sequencing. The obtained sequences were analyzed using Genius Prime 2020 (Biomatters, Auckland, New Zealand) software and compared to available 16S rRNA sequences in the GenBank database using the BLASTn program (Basic Local Alignment Search Tools).

### 2.5. In vitro Bacterial Screening for Plant Growth-Promoting Activities 

#### 2.5.1. Bacterial Phosphate Solubilization in Liquid Medium

The isolated bacterial strains were assessed for their ability to solubilize hydroxyapatite (TCP) and polyphosphate fertilizer (PolyP) in liquid National Botanical Research Institute phosphate growth medium (NBRIP) media separately [31], containing 5 g and 4.9 g of each as the sole P source. Then, the pH was adjusted to 7 before autoclaving. Briefly, 100 µL of an overnight culture of each bacterial suspension (OD_600nm_ = 0.8) (UV/visible spectrophotometer, UV-6300PC) was added to 50 mL of NBRIP broth and then incubated at 28 °C at 140 rpm (New Brunswick™ Innova^®^ 43 Incubator Shaker Series). As a negative control, we used non-inoculated NBRIP media. After 7 days of incubation, 1 mL of each culture was collected and centrifuged at 10,000 rpm for 10 min before filtration using 0.22 µm filters (Merck KGaA, Darmstadt, Germany). Soluble P was determined using an Inductively Coupled Plasma Optical Emission Spectrometer (ICP-OES) [32]. The final pH of bacterial culture supernatants was recorded using a Fisherbrand™ pH meter. As a positive control, we used the strain *Bacillus licheniformis* QA1, known for its ability to solubilize hydroxyapatite [24]. For statistical analysis, each experiment was carried out in triplicate, and we subtracted the control values from the results to calculate the P solubilization capacity. 

#### 2.5.2. Bacterial Potassium Solubilization 

Potassium (K) solubilization was first qualitatively assessed on plates containing the Alexandrov agar medium (containing per L: 5 g glucose, 0.5 g MgSO_4_·7H_2_O, 0.1 g CaCO_3_, 0.006 g FeCl_3_, 2 g KH_2_PO_4_, and 5 g of mica as an insoluble source of potassium) [33], and using mica (2.35% CK; 2.15% NK; 45.94% OK; 1.07% NaK; 1.13% MgK; 19.45% AlK; 18.22% SiK; 0.04% PK; 0.50% SK; 7.71% KK; and 1.44% CaK) as the sole source of K. Then, 10 µL of each strain was inoculated into Alexandrov agar plates and incubated at 28 °C for 11 days [34]. 

Subsequently, to provide a more accurate and reproducible assessment of the solubilization ability of bacterial strains, allowing for a more reliable comparison of solubilization capabilities, we tested the ability of only three bacterial strains (chosen for the in vivo experimentation) to solubilize K in liquid media. The K solubilization rate was quantitatively monitored in Alexandrov liquid media containing mica. One mL (OD_600nm_ = 0.8) of a 48-h culture of each strain was added to 100 mL of Alexandrov broth and incubated at 28 °C under shacking at 140 rpm (New Brunswick™ Innova^®^ 43 Incubator Shaker Series). Subsequently, 7 days post incubation, the medium was filtered using 0.22 µL filters (Merck KGaA, Darmstadt, Germany), and the K release in inoculated and non-inoculated treatments was measured by atomic absorption spectrometry at the Agriculture Innovation and Technology Transfer Center (AITTC) of UM6P, Benguerir, Morocco. In parallel, we recorded the final pH of each supernatant. To emphasize the bacterial activity for the statistical analyses, we subtracted the negative control values from our results. 

#### 2.5.3. Bacterial Zinc Solubilization 

The ability of bacteria to solubilize insoluble forms of Zn was evaluated in Tris-mineral agar medium amended with Zn oxide (ZnO) (1.244 g/L = 15.23 mM), and Zn carbonate (ZnCO_3_) (1.728 g/L = 5.2 mM) at a 0.1% Zn final concentration. The Tris-mineral agar medium contains D-glucose: 10.0 g/L; (NH_4_)_2_SO_4_: 1.0 g/L; KCl: 0.2 g/L; K_2_HPO_4_: 0.1 g/L; and MgSO_4_: 0.2 g/L. The pH was adjusted to 7.00 ± 0.25 before autoclaving [35]. Single colonies were spot-inoculated on plates and incubated for 10 days at 28 °C. Zn solubilization correlates with the appearance of halo zones around the colonies [36]. Zinc solubilization was linked to the presence of a clear zone around colonies. Efficiency was categorized as absent (−), low (+), moderate (++), and strong (+++).

#### 2.5.4. Bacterial Indole-3-Acetic Acid (IAA) Production 

The amount of IAA (indole-3-acetic acid) produced by PSB strains was measured by cultivating them in Trypticase soy broth (TSB) broth, containing 0.1% L-tryptophan, which acts as a precursor for IAA synthesis [37]. Bacteria were grown in 50 mL of the prepared medium, kept at 28 ± 2 °C, and shaken at 150 rpm for 7 days [38]. Then, row bacteria were spun at 12,000 rpm for 10 min at 4 °C. The supernatants were then filtered through 0.22 µm sterile syringe filters. Then, Van Urk Salkowski reagent, made up of 1 mL of 35% HClO_4_ and 0.5 M FeCl_3_, was mixed with 1 mL of each culture filtrate. The mixture was left in the dark for 30 min at room temperature. The production of IAA was monitored by the appearance of a pinkish color. The absorbance was then measured at 535 nm using a VWR^®^ UV-6300PC spectrophotometer. The concentration of IAA produced was determined using a standard curve made with pure IAA (Sigma Aldrich, Overijse, Belgium) in the range of 0 to 100 g/mL. For statistical analyses, we subtracted the negative control values from each tested sample to emphasize the bacterial activity.

#### 2.5.5. Bacterial Ammonia Production 

The production of ammonia by the isolated PSB strains was evaluated using the method described by [39]. Essentially, 100 µL (OD_600nm_ = 0.8) of each bacterial suspension was added to tubes containing 10 mL of peptone water, incubated at 28 °C, and shaken at 150 rpm for 96 h. No inoculated medium was used as a negative control. Then, 1 mL from each sample was centrifuged at 10,000 rpm for 10 min. Subsequently, 0.5 mL of Nessler’s reagent was added to each supernatant [39]. Ammonia production was considered positive when a brownish coloration appeared, and the absorbance was measured at 450 nm (VWR^®^ UV-6300PC, spectrophotometer). Finally, ammonia concentrations were calculated using a standard curve of ammonium sulfate ranging from 0 to 0.3 µmol/mL [40]. As a positive control, we used *Pseudomonas frederiksbergensis* (S6 C+) [41]. To highlight the bacterial activities, we subtracted the values obtained in the negative control for statistical analyses. 

### 2.6. In vitro Screening for Biocontrol Activities

#### 2.6.1. Bacterial Siderophores Production

Siderophores production by bacteria was monitored using a Chrome Azurol S (CAS) agar assay [42]. The study used: (1) Fe-CAS indicator solution, prepared by dissolving 0.06 g of CAS in 50 mL of distilled water and mixing it with 10 mL of iron solution (10 mM HCl, 1 mM FeCl_3_.6H_2_O). Under shaking, we added 0.073 g of HDTMA (hexadecyltrimethylammonium bromide) dissolved in 40 mL of distilled water. The obtained blue solution was then autoclaved; (2) mixing a salt solution containing 100 mL of distilled water, 3 g of KH_2_PO_4_, 5 g of NaCl, and 10 g of NH_4_Cl, and adjusting the pH to 6.8. (3) Buffer solution, prepared by dissolving 32.24 g of PIPES (Pipérazine-N, N′-bis (2-éthanesulfonique) in 7.5% of solution 2 after adding 15 g of agar, solution (3) was autoclaved; and (4) casamino acid solution, prepared by dissolving 3 g of casamino acid in 27 mL of distilled water, and sterilized using 0.22 µm filters. All solutions were mixed and poured on plates that were spot inoculated with bacteria and incubated for 7 days at 28°C. The bacteria that produce siderophores form a halo around colonies due to iron chelation. The results were visually examined based on the width of the halo observed around the blue medium. The effectiveness levels were categorized as absent (−), low (+), moderate (++), and strong (+++).

#### 2.6.2. Bacterial Hydrogen Cyanide Production

The production of hydrogen cyanide (HCN) by bacteria was monitored as previously described [43]. The process involved adding 0.44% glycine to Tryptic Soy Agar (TSA) medium, then spreading 100 µL (OD_600nm_ = 0.6) of each bacterial strain on poured agar plates using a sterilized glass spreader. A filter paper was placed under the lid of the plates after being soaked in 0.5% picric acid in 2% sodium carbonate for 1 minute. The Petri dishes were locked with parafilm and incubated at 28 °C for 96 h. The production of HCN was tracked using Whatman paper, with the color of the paper changing from yellow, caused by sodium picrate solution, to orange or brown. The intensity of production was determined visually. The effectiveness levels ranged from absent (−) to low (+), moderate (++), and strong (+++).

#### 2.6.3. Bacterial Antifungal Activity

The bacterial strains were investigated in an in vitro assay to determine their potential antagonism activities against two different Fusarium isolates: *F. culmorum* sucardu and *F. culmorum* P.V (*n* = 3). Bacterial cells from a fresh overnight culture were streaked in a loop on PDA plates (Biokar Diagnostics, Allonne, France). The next step was to position a plug of actively growing mycelium at the same separation from the bacterial line. Plates containing only fungus were used as a control. The dishes were covered with parafilm and kept at 28 °C for 4 days. The development of fungi was monitored daily.

### 2.7. Assessment of Strains’ Activities on Seeds

#### 2.7.1. Germination Assay

Bread wheat (*Triticum aestivum* cv. Emperator) seeds were utilized for in vitro seed bacterization to assess the impact of the isolated bacteria on early plant development. Firstly, seeds were sorted, and any seeds showing signs of damage were removed from the selection. Seeds were surface sterilized using 98% ethanol (30 s) and 2% sodium hypochlorite (2 min), successively washed three times with distilled sterile water, and air-dried under a laminar flow hood. Then, seeds were inoculated by drenching in a bacterial suspension containing 10^10^ CFU/ml of log phase culture in magnesium sulfate-buffered solution (MgSO_4_, 10 mM) for 1 h with 30 rpm shaking on a HS 501 digital agitator (Fisher Scientific SAS, Illkirch Cedex, France) [44]. The control seeds were drenched in MgSO_4_ solution only, in the absence of bacteria. Afterwards, under a laminar flow hood, 10 seeds were placed in sterile Petri dishes (100 mm × 15 mm) containing sterile Whatman paper and moistened with 3 mL of sterile distilled water. Plates were initially incubated at 28 °C for 48 h in the dark and were subsequently kept for 7 days at room temperature in a day/night cycle (~12/12 h). The percentage of germinated seeds was determined after 48 h using Equation (1). Seven days post-incubation, the vigor index was calculated using Equation (2):Germination rate (%) = (*n* ÷ N) × 100(1)
where “*n*” is the number of germinated seeds and “N” represents the total number of seeds.
Vigor index = % germination rate × total plant length(2)

The experiment was designed according to a complete randomized block design with three replicates and nine treatments: (1) control without seed inoculation and (2) to (8) treatment with strains P1 to P8. 

#### 2.7.2. Assessment of Bacterial Activities on Plants

The soil used in this experiment was collected from a depth of 0–20 cm from a field (50°23′27.5″ N 4°02′39.4″ E) previously sown with maize in the province of Hainaut, Belgium. This soil was characterized as clay-loamy with a reduced content of available P (2.3%), 14 of K (%), 0.05 of total N (%), 3554.9 of Ca (%), 8.2 of pH KCL, 19 of Mg (%), and 2.4 of organic matter (g/kg). Wheat (*Triticum aestivum* variety Emperor) was grown in pots (30 cm in height, 11 cm in diameter) containing 2.3 kg of a mixture of P-characterized soil and sand (2:1 *w*/*w*). The experimentation was carried out in a growth chamber at Gembloux-Agro-Bio-Tech, Belgium. Pots (five seeds per pot) were arranged according to a completely randomized block design with five pots per treatment. The 16 h photoperiod was applied with an average photosynthetic photon flux density (PPFD) of almost 240 μmol m^−2^ s^−1^ at 23 °C with 60% relative humidity. Fertilization treatment was performed using PolyP fertilizer. The amount of fertilization needed was calculated based on the soil analysis results and bread wheat’s nutritional needs. Wheat was fertilized with N, P, and K at 180, 60, and 150 mg/kg of dry soil, respectively. 

#### 2.7.3. Biomass and Chlorophyll Index Measurements

At the growth stage Z69, wheat was harvested and divided into shoots and roots, cleaned, and oven-dried at 75 °C until stabilization. The chlorophyll content index (CCI) is essential in yielding as it significantly and positively correlates with the grain yield and harvest index [45]. CCI was measured using a non-destructive portable chlorophyll meter (MC-100 Chlorophyll Concentration Meter, Apogee Instruments). Five measurements were carried out at different wheat growth stages: Z22, Z29, Z32, Z37, and Z45, corresponding to tillering beginning, first node detection, stem elongation, heading, and flowering, respectively [28]. In each pot, three leaves were marked for successive measurements. The machine gives the mean of three measurements on each leaf for more precision. 

### 2.8. Statistical Analysis 

The in vitro tests were performed in triplicate, while the in vivo experiments were performed in five replicates. Using IBM SPSS Statistics 20 software, all acquired information was statistically analyzed. Analysis of variance (ANOVA) was followed by a post-hoc evaluation with Tukey’s Studentized Range (HSD) test. A *p* value < 0.05 was deemed statistically significant. The data present the means ± Standard Error of the Mean (SEM).

## 3. Results 

### 3.1. Bacterial Isolates P1 to P8 belong to the Genera of Pseudomonas or Bacillus 

Eight colony types with different morphologies that could be distinguished on the plates were re-isolated, purified, and designated P1 to P8. Based on sequencing the gene-encoding 16S rRNA of the eight isolates P1 to P8, it was revealed that five of the isolated strains corresponded to the *Pseudomonas* genus: P1 matched with *P. soyae* with 99.3% identity, P2 and P7 corresponded to *P. reinekei* with 98.9% and 99.2% identities, respectively, P3 shared a 99.2% identity with *P. koreensis*, and P8 shared 98.7% with *P. vancouverenesis*. Two isolates corresponded to the *Bacillus* genus; P4 exhibited 99.7% identity to *B. tropicus*, and P5 corresponded to *B. paramycoides* with 99.4% identity. Lastly, P6 was 99.3% identical to *Peribacillus frigoritolerans*. The 16S rDNA sequences were deposited in GenBank with accession numbers OR724991 to OR725006.

### 3.2. Bacterial Isolates P1, P5, and P6 Solubilize Both Insoluble Forms of P

The quantification of released available P in a liquid NBRIP medium revealed the ability of P1, P5, and P6 isolates to solubilize both forms of insoluble P. In contrast, P4 and P8 isolates solubilized only PolyP. Furthermore, P8 solubilized hydroxyapatite but not PolyP (Figure 1). Compared to the positive control *B. licheniformis* QA1 strain, hydroxyapatite rate solubilization by P6 was 666.025 ± 0.17 mg/L, while P8 showed the highest solubilization, reaching up to 1063.45 ± 3.6 mg/L (Figure 1). The inoculation by bacteria under the application of both P sources showed a negative correlation between the quantities of soluble phosphate and pH values; r = −0.66 and r = −0.74 for hydroxyapatite (TCP) and PolyP, respectively.

### 3.3. Bacterial Isolates P1, P2, and P4 Solubilized Potassium

Potassium solubilizing in a solid medium showed that only P1 isolate solubilized potassium. However, in the liquid medium, all three tested isolates (P1, P2, and P4) could solubilize potassium with values ranging from 6.1 ± 0.2 to 13.7 ± 0.3 mg/L **(**Figure 2). The pH variation between the different cultures inoculated with strains P1, P2, or P4 was not significant. Compared to the control, the bacteria did not acidify the medium. 

### 3.4. P1, P2, P3, P7, and P8 Isolates Solubilized Both ZnO and ZnCO_3_, While P4 Solubilized Only ZnO

Here, we demonstrated that isolates P1, P2, P3, P7, and P8 solubilize both ZnO and ZnCO_3_ insoluble forms (Table 1). In contrast, P4 solubilized only ZnO, while P5 and P6 isolates did not solubilize any tested form of Zn. 

### 3.5. Bacillus Tropicus P5 Strain Is the Highest Ammonia Producer

All tested bacteria produced ammonia in significantly higher concentrations compared to the positive control, *P. frederiksbergensis* (strain S6) [41] (Table 1). Depending on the amount of ammonium production, there are four possible groupings for the results. The first group includes the highest value of 9.1 ± 0.44 µmol/mL, which was produced by *B. tropicus* P5. The second group includes *P. reinekei* P2, *P. frigolitolerans*, *P. reinekei* P7, and *P. vancouverenesis* P8, with ammonia production values ranging from 5.1 ± 0.2 to 5.6 ± 0.1 µmol/mL. The third group includes *P. soyae* P1 and *B. tropicus* P4, producing 3.3 ± 0.1 and 3.01 ± 0.1 µmol/mL, respectively. The last group contains *P. koreensis* P3, with a production of 1.7 ± 0.1 µmol/mL. 

### 3.6. Production of IAA Is Highly Pronounced in P. soyae P1 and B. tropicus P4 Strains 

Following seven days of incubation in TSB media supplemented with 0.1% L-Tryptophan as a precursor, we demonstrated that all strains produced IAA, with a significantly higher amount obtained with *P. soyae* P1 reaching 44.1 ± 2 µg/mL, while the lowest concentration was 7.5 ± 0.3 µg/mL, expressed by *B. tropicus* P4 (Table 1). 

### 3.7. HCN Are Produced by All Strains 

HCN exhibit different plant benefits, mainly by functioning as metabolic inhibitors against phytopathogens. Except for the lower performance of *P. frigotolerans* P6 and *P. reinekei* P7; the other isolates produce a significant amount of HCN (Table 2 and Figure 3C). 

### 3.8. Siderophores Production Was Higher in the P. reinekei P2 Strain 

The *P. reinekei* P2 strain was the best siderophores producer (Table 2 and Figure 3B). The results were visually analyzed regarding the halo width against the blue medium, and the experiment was performed in triplicate. 

### 3.9. Pseudomonas reinekei P2 Strain Exhibited Stronger Anti-Fusarium Activity

We investigated the antifungal activity of the studied bacteria against two *Fusarium culmorum* isolates (sucardu and PV). Even though all our strains limited the development of both fungal isolates, various phenotypes were observed (Table 3 and Figure 4). The behavior of the tested bacteria toward fungi can be tentatively divided into three classes. The first is defined by the total inhibitory effect of *F. culmorum* sucardu development zones exhibited mainly by *P. soyae* P1 and *P. reinekei* P2 strains. The second class with a limited fungal development zone was detected using strains *B. paramycoides* P5 and *Peribacillus frigoritolerans* P6 (Figure 4). The third class is represented by the antifungal behavior of *B. paramycoides* P5 and *Peribacillus frigoritolerans* P6 against *F. culmorum* PV, which is more pronounced compared to *F. culmorum* sucardu.

### 3.10. P. frigoritolerans P6 and B. tropicus P4 Strain Optimally Enhanced Wheat Seeds’ Germination 

To enhance global agricultural output and maximize nutrient-use effectiveness, PGPR are occasionally used to inoculate seeds, soil, or plants (Alori et al., 2017 [16]). Here, inoculating bread wheat seeds revealed that *Peribacillus frigoritolerans* P6 significantly increased both the germination rate, reaching up to 97.43 ± 3.41% (Table 3), and the plant vigor index (Figure 5). Compared to the control, seeds treated with *P. frigoritolerans* P6 culture suspension improved the germination rate by 3.2 times. Similarly, the length of seedlings was increased by up to 1.39 times (Figure 6). The strain *B. tropicus* P4 increased the root length by 1.46 times, while the germination rate reached up to 2.83 times (Table 3). The strain *P. vancouverensis* P8 also improved the germination rate by up to 2.8 times.

The bacterial suspensions impact the germination rate of bread wheat seeds in vitro. The numbers show means and standard deviations for replicates (*n* = 3).

### 3.11. Co-Application of PolyP and P. soyae P1 Strain Boost Shoot Biomass While the Sole Application of PolyP Induced Root Development

Based on in vitro complementary results (P hydrolyzation/solubilization, Zn solubilization, IAA production, HCN, and siderophore production), which suggest their beneficial potential for plant growth and biocontrol activity, three strains (*P. soyae* P1, *P. reinekei* P2, and *B. tropicus* P4) were selected for further study. We then investigated the in vivo efficiency of the bacterial suspension in the growth-controlled chamber. Before sowing, seeds were sterilized, and then the three selected strains were applied individually or in co-application with PolyP. Eight treatments were used: (1) (C−) negative control (no bacterial nor PolyP); (2) (C+) positive control (PolyP); (3) seeds treated with strain *P. soyae* P1; (4) co-application of *P. soyae* P1 and PolyP; (5) seeds treated with *P. reinekei* P2 strain; (6) co-application of *P. reinekei* P2 and PolyP; (7) seeds treated with strain *B. tropicus* P4; (8) co-application of *B. tropicus* P4 strain and PolyP. Shoot (Figure 7A) and root (Figure 7B) biomass were measured. Our results revealed that the combination of *P. soyae* P1 and PolyP showed the highest shoot biomass, reaching up to 41.99 g ± 0.87 g, although not significantly different from five other treatments (Figure 7A). Remarkably, the application of *P. soyae* P1, *B. tropicus* P4 alone gives similar results as the application of PolyP alone (Figure 7A). Interestingly, no significant differences were observed between the application of PolyP alone or when *P. soyae* P1, *B. tropicus* P4, and *P. reineki* P2 were applied individually or combined with PolyP (Figure 7A). 

As for root biomass, treatment with PolyP alone increased root biomass followed by the *B. tropicus* P4 strain and their combination (Figure 7B). Compared to the control, no significant change in root development was detected in the other treatments (Figure 7B). 

To further investigate the impact of bacterial inoculation and the combination of bacterial suspension and PolyP treatment on bread wheat growth stages, we determined the leaf chlorophyll content index (CCI). At the Z32 stem elongation stage, wheat inoculated with the co-application of *P. reinekei* P2 and PolyP showed a significant increase in the CCI compared to both the control and the wheat fertilized with PolyP alone. The CCI improved from 24.12 µmol/m^2^ ± 5.45 and 24.18 µmol/m^2^ ± 4.41 in the control and in wheat under PolyP fertilization, respectively, while the CCI reached 37.7 µmol/m^2^ ± 4.18 in wheat under the co-application of *P. reinekei* P2 and PolyP (Figure 7C). Remarkably, at the Z37 heading growth stage, a pronounced improvement in the CCI value was detected in wheat treated with the co-application of B. tropicus P4 and PolyP (37.02 µmol/m^2^ ± 3.28) compared to wheat fertilized with PolyP alone (24.07 µmol/m^2^ ± 1.08) and the control (23.06 µmol/m^2^ ± 1.53) (Figure 7C).

Overall, the isolated strains provided similar or even better results than PolyP in shoot biomass and chlorophyll content globally. Among the combination of PolyP and the isolated strains, the co-application of *P. reinekei* P2 or *B. tropicus* P4 showed significantly higher efficacy at Z32 or Z37, respectively. All treatments improved shoot biomass compared to the control, while only PolyP and *B. tropicus* P4 improved root biomass. PolyP was the most efficient for promoting root growth, while the *B. tropicus* P4 strain was the only strain offering a significant difference in comparison with the control.

## 4. Discussion

Technological advances in utilizing rhizosphere microbes are anticipated to enhance plant nutrition and soil health, potentially leading to improved crop yields and sustainability. There is a growing interest in biostimulants to enhance plant growth and development and to improve plant nutrition and soil health. On the other hand, PolyP fertilizers have been of great interest in agriculture thanks to their potential to improve efficient phosphorus (P) use and crop productivity. Several studies have investigated the impact of PolyP fertilizers on soil physicochemical properties, root-microbial activities, and crop physiology, highlighting their potential benefits [4,26,46]. Nevertheless, the co-application of PolyP and potential biostimulant bacterial strains has never been studied.

### 4.1. Solubilization of P, K, and Zn by P1-P8 Strains

Bacteria use various molecular mechanisms to solubilize/hydrolyze insoluble P. Here, we demonstrated that *P. soyae* P1, *B. paramycoides* P5, and *P. frigoritolerans* P6 strains solubilized TCP and hydrolyzed PolyP in vitro. Our findings also show that the solubilization of PolyP was accompanied by an acidification of the medium, with pH values reaching up to 4. These results are in agreement with a recent study carried out in liquid NBRIP medium with other strains [47]. Several studies have hypothesized that soil microbes can be involved in PolyP hydrolysis, yet the mechanisms used by PSB to hydrolyze PolyP remain poorly described [9,48,49]. Some bacteria solubilize mineral phosphate by directly oxidizing glucose to gluconic acid, which chelates the mineral phosphate and makes it more available to plants [16]. Of all the organic acids, gluconic acid is the most frequent agent of mineral phosphate solubilization, as it chelates the mineral phosphate and makes it more available to plants [16]. Other bacteria liberate enzymes (including non-specific acid phosphatases (NSAPs); phosphatases and phytase) that act on organic compounds to produce acids that solubilize phosphorus (Othman and Panhwar 2014 [50]; Alori et al. 2017 [16]). 

In addition to PSB bacteria, potassium solubilizing bacteria (KSB) convert the insoluble minerals K into soluble forms available for plants [51]. KSB use several mechanisms including chelation, acidolysis, the lowering of pH, exchange reactions, and complexation [52]. To make potassium available for crop production, biological solutions involving KSM have been proposed [53,54,55]. Here, we found that *P. soyae* P1, *P. reinekei* P2, and *B. tropicus* P4 can solubilize K in liquid media. However, in solid media, K solubilization was only detected for the *P. soyae* P1 strain. Potassium solubilization was not accompanied by medium acidification. KSB like *Pseudomonas* and *Penicillium* can solubilize potassium without significantly altering the soil pH. This process involves organic acid production, siderophore synthesis, and extracellular polymeric substance formation. 

Zinc is mostly found in the soil in molecules such as smithsonite (ZnCO_3_), sphalerite (ZnS), zincite (ZnO), franklinite (ZnFe_2_O_4_), willemite (Zn_2_SiO_4_), and hopeite (Zn_3_(PO_4_)_2_·4H_2_O), all of which are insoluble and unusable by plants. Low zinc availability hinders crop growth and reduces zinc levels in seeds and grains, affecting nutritional quality and leading to zinc deficiency in humans [56]. Zinc (Zn) is a rare element necessary for several plant biological activities, including enzyme activation and protein synthesis. Zn shortage occurs in calcareous, sodic, sandy, and intensively farmed soils, affecting the yield of crops including maize, rice, potatoes, wheat, and tomatoes [57]. Soil microorganisms like *Pseudomonas*, *Bacillus*, and *Burkholderia* can solubilize zinc, enhancing its availability and uptake by plants, making it a viable alternative to chemical fertilizers [58,59]. ZSB use acidification, siderophores, and oxidoreductive systems to solubilize zinc, sequester cations, and enhance solubility by producing organic acids in soil [60]. Here, we demonstrated that all tested *Pseudomonas* strains, *P. soyae* P1, *P. reinekei* P2, *P. koreensis* P3, *P. reinekei* P7, and *P. vancouverensis* P8, solubilized ZnO and ZnCO_3_.

### 4.2. Strains’ Biocontrol Activity against Fusarium Culmorum

*Pseudomonas* species are among the most common PGPR. Recently, we reported that the class of Pseudomonadota (formerly known as Proteobacteria) was highly abundant in the wheat rhizosphere microbiome amended with the same PolyP fertilizer [26]. *Pseudomonas* strains have been proposed as biocontrol agents and plant growth promoters (PGP) due to their ability to suppress pathogenic microorganisms, synthesize growth-stimulating plant hormones, and promote plant growth [61,62]. Beyond their PGP properties, we demonstrated that strains *P. soyae* P1 and *P. reinekei* P2 completely stopped the development of *F. culmorum* sucardu. This activity might be mediated by siderophores that stimulate the production of metabolites involved in phytopathogen control [63]. The hypothesis suggests that siderophores, by regulating iron bioavailability and exhibiting antimicrobial properties, can negatively impact various phytopathogens. The presence of siderophores seems to stimulate the production of metabolites that contribute to the control of phytopathogens, highlighting the intricate role of siderophores in plant-microbe interactions and disease management in agriculture [64,65]. Strains *P. soyae* P1 and *P. reinekei* P2 totally limit the development of *F. culmorum* sucardu without totally inhibiting the growth of *F. culmorum* P.V. This difference is likely associated with various factors, including the specificity of the inhibitory mechanisms and the genetic and biochemical characteristics involved in the interacting microorganisms.

Furthermore, we observed that the *B. tropicus* P4 strain produces siderophores and exhibits remarkable antifungal activity against both *F. culmorum* sucardu and *F. culmorum* PV. *B. tropicus* has already been identified as a potential plant growth-promoting bacterium [66,67,68] and biocontrol agent against plant pathogens [69]. 

Furthermore, each isolated strain from our study produced a high amount of HCN. It is well known that some bacteria, like endophytic *Streptomyces* species, produce HCN, which functions as an antifungal agent and suppresses *Fusarium* disease [70]. Additionally, the synthesis of siderophores, HCN, and volatile antifungal chemicals by endophytic actinobacteria has been emphasized as an antagonist compound against *Fusarium* species [71]. 

Bacterial ability to inhibit a specific pathogen is often determined by their production of antimicrobial compounds, competition for resources, and other ecological interactions. Moreover, the outcome of microbial interactions can be influenced by their genetic and biochemical characteristics, such as virulence factors, resistance mechanisms, and metabolic pathways. The complexity of microbial communities and their interactions further contributes to the differential inhibition of pathogens by bacterial strains, as the presence of other microorganisms and environmental factors can modulate the outcome of these interactions. Understanding these dynamics could pave the way for new strategies in pathogen control, leveraging the natural capabilities of beneficial bacteria to enhance plant health and productivity. Future research could focus on identifying key genetic and biochemical traits that confer inhibitory abilities, as well as optimizing environmental conditions to maximize the efficacy of biocontrol agents. By combining biocontrol properties that help suppress plant pathogens with the ability to solubilize phosphorus and enhance nutrient availability for plants, these dual-function strains can provide a comprehensive solution for improving plant health and sustainable productivity while reducing the reliance on chemical inputs. The research and application of such strains aim to optimize plant growth, protect against diseases, and enhance nutrient uptake efficiency in agricultural systems, contributing to environmentally friendly and economically viable farming practices.

### 4.3. Impact of PolyP-PSB Co-Application on Wheat Growth Parameters

We recently reported that the treatment of wheat plants with PolyP increased wheat growth [26]. Here, we confirmed PolyP’s positive effect on wheat growth and revealed the co-application of *P. soyae* P1 and PolyP’s positive impact on shoot biomass. Additionally, *B. tropicus* P4 enhanced both root and shoot biomasses. Shoot chlorophyll II content was linked with shoot biomass in late but not early growth stages. This can be explained by plant P absorption improvement, as chlorophyll fluorescence measurement forecasts photosynthetic process effectiveness [72,73]. As expected, with the soil poor in P, the applied PolyP and PSB strains enhanced the aboveground characteristics of the wheat compared to the control. Interestingly, the co-application of *B. tropicus* P4 and PolyP induced the highest chlorophyll content at Z37. The genome sequence analysis of *B. tropicus* identified several key plant-growth-promoting genes, including those encoding the auxin efflux carrier protein, cytokinin 9-b-glucosyltransferase, and alkaline phosphatase, which boost plant growth and development and improve phosphate solubilization [74,75,76,77]. In addition, this genome also harbors genus-specific siderophores biosynthesis proteins such as bacillibactin and anthrachelin, essential for iron acquisition in limiting environmental conditions and for biocontrol [67].

Given that *P. soyae* P1, *P. reinekei* P2, and *B. tropicus* P4 demonstrate PGPR traits, we explored their impact on wheat growth and chlorophyll content. PGPR can significantly enhance plant growth, particularly through the production of IAA, which promotes root growth and branching, leading to an extensive root system that allows the wheat plant to better explore the soil and access more nutrients and water [78,79]. This was evident in our in planta study where wheat treated with *B. tropicus* P4, used for its high IAA production, exhibited increased root biomass compared to the control. Furthermore, *P. soyae* P1 and *B. tropicus* P4 strains demonstrated the ability to solubilize/hydrolyze P, converting insoluble phosphates into soluble forms. This action improves phosphorus availability for wheat plants, thereby enhancing nutrient uptake and promoting better tillering [80]. This was substantiated by the increased shoot biomass in wheat treated with strains *P. soyae* P1 and *B. tropicus* P4, compared to the control. Additionally, the bacterial production of ammonia enriches soil nitrogen levels, promoting robust plant growth, greener foliage, and a higher protein content in wheat grains [81]. Ammonia contributes to chlorophyll formation, thereby enhancing photosynthesis and overall plant health [82]. This effect was observed in the chlorophyll content index of wheat treated with *P. soyae* P1, *P. reinekei* P2, or *B. tropicus* P4, which showed significant improvement compared to the control.

The individual applications of PolyP and PSB showed comparable effects on both root and shoot biomasses, indicating that either treatment alone is effective in promoting plant growth. However, when PolyP and PSB were applied together, the combined treatment did not result in any significantly enhanced growth or improvement in root and shoot biomasses compared to the individual treatments. This suggests that, at this stage of the study, the co-application of PolyP and PSB does not provide any additional benefits beyond what is achieved by using either PolyP or PSB alone. Further research might be needed to explore potential additional effects under different conditions or at different growth stages. Although the different bacterial strains exhibited overall similar effects on plant growth, minor differences were observed between them. These variations could be due to specific interactions between the strains and wheat plants or other environmental factors. To better understand these differences, if any, and their potential implications, further investigations are required. Additionally, it would be valuable to evaluate the strains’s effectiveness as PSB and biocontrol agents in vivo. This could help determine whether there are additional effects that could enhance plant growth and health more effectively than the use of PSB or biocontrol agents alone.

Overall, the results do not conclusively state that observing one trait always indicates the presence of another for a given PGPR. The traits seem to be linked, but the strength and consistency of these links are not quantified. Additionally, while some traits may be more common within certain genera, the results do not provide enough information to determine if the same specific combination of traits is always observed for a given genus or species of PGPR. In summary, the results suggest a connection between various PGPR properties, but more targeted research would be needed to determine the consistency and specificity of these links across different PGPR genera and species.

## 5. Conclusions

In this study, we focused on three strains, namely *P. soyae* P1, P. *reinekei* P2, and *B. tropicus* P4, selected based on their in vitro plant growth-promoting properties. Our findings indicated that the selected bacterial strains also exhibited remarkable biocontrol activity against *F. culmorum* strains, suggesting their potential application as potent plant growth-promoting bacteria (PGPB) with biocontrol capabilities. These strains were further evaluated in vivo using wheat grown on P-deficient soil in a controlled chamber to determine their impact on wheat growth parameters. Specifically, we investigated whether the combined application of PolyP to each of the three strains would have additional effects on wheat growth. Nonetheless, the outcomes do not appear to be particularly encouraging in terms of improving crop growth by combining PolyP and one of the strains. However, our results lay the basis for further experiments to better understand the observed effects. Future studies should focus on optimizing this finding through additional in vivo experiments, aiming for a more comprehensive understanding of the interactions between PolyP and specific bacterial strains in promoting wheat growth. This includes delineating the exact mechanisms by which PolyP influences bacterial activity for both PBP and biocontrol traits, quantifying the effects on wheat growth under various environmental conditions, and identifying the most effective bacterial strains for enhanced P solubilization, uptake, and biocontrol abilities.

## Figures and Tables

**Figure 1 microorganisms-12-01423-f001:**
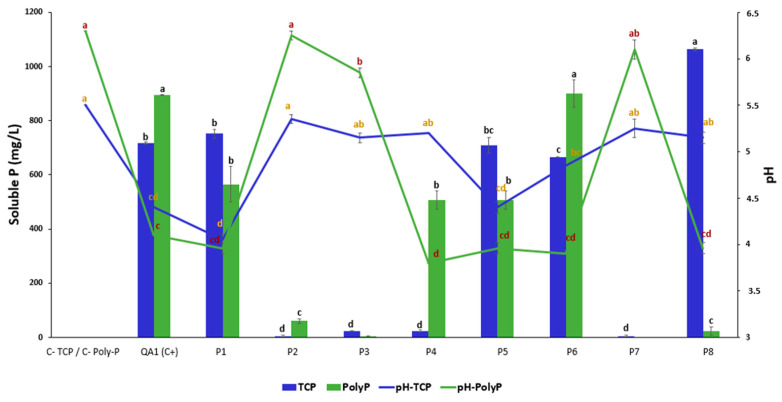
Phosphate solubilization/hydrolyzation by the isolated bacteria in the presence of 0.5% hydroxyapatite (TCP) and polyphosphate in NBRIP broth. The green and blue lines indicate pH variations according to bacterial isolates P1 to P8 and the nature of P forms. Strain *B. licheniformis* QA1 was used as a positive control. The various letters in superscript (a, b, c, and d) denote the statistically significant difference between treatments at a 95% confidence level.

**Figure 2 microorganisms-12-01423-f002:**
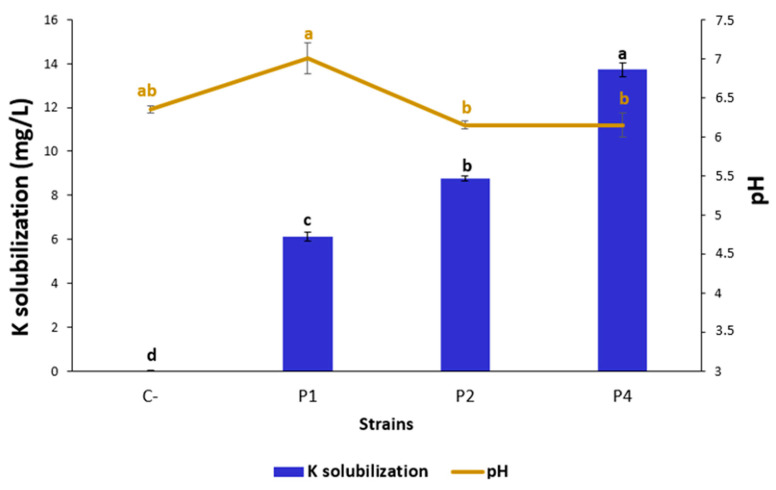
Strains’ ability to solubilize potassium. The figure represents the means of three replicates. The negative control is represented by non-inoculated media (C-). Testing was conducted on three strains (P1, P2, and P4) selected for in vivo experimentation. Reporter letters in superscript (a, b, c …) represent the 95% statistically significant difference between treatments.

**Figure 3 microorganisms-12-01423-f003:**
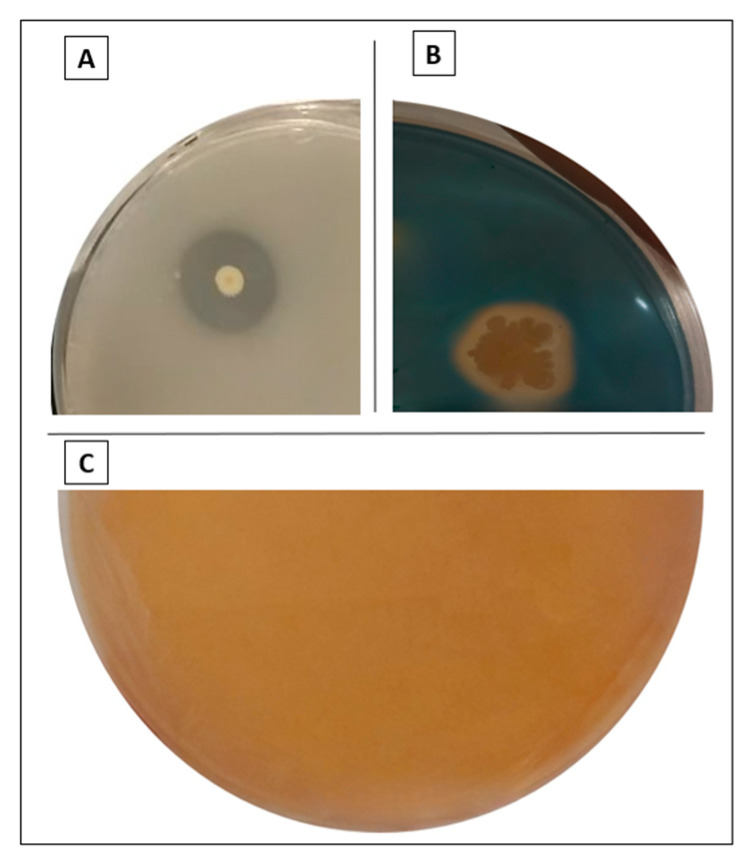
Illustration of some plant growth-promoting properties of some isolated strains. (**A**) zinc solubilization (*P. koreensis* P3 solubilizing ZnCO_3_), (**B**) siderophores production (*P. reinekei* P2) and (**C**) HCN production (*P. soyae* P1).

**Figure 4 microorganisms-12-01423-f004:**
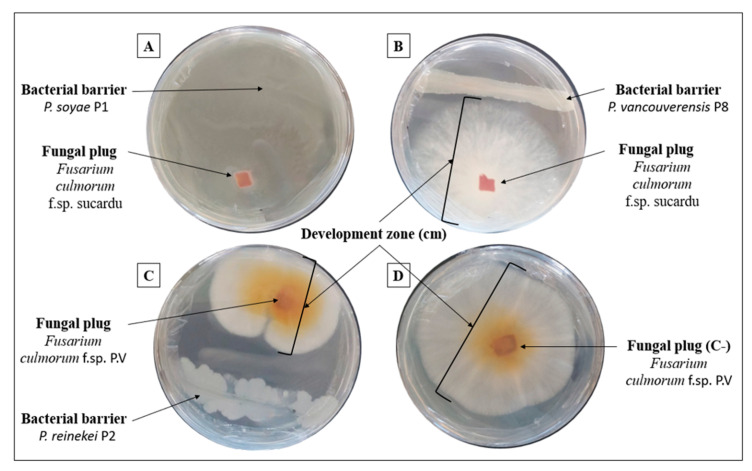
Illustration of the in vitro antagonism assay. Strains were incubated with *Fusarium culmorum* sucardu (**A**,**B**) and *Fusarium culmorum* P.V (**C**,**D**) using potato dextrose agar (PDA) plates. *F. culmorum* P.V. Control in the absence of bacteria (**D**).

**Figure 5 microorganisms-12-01423-f005:**
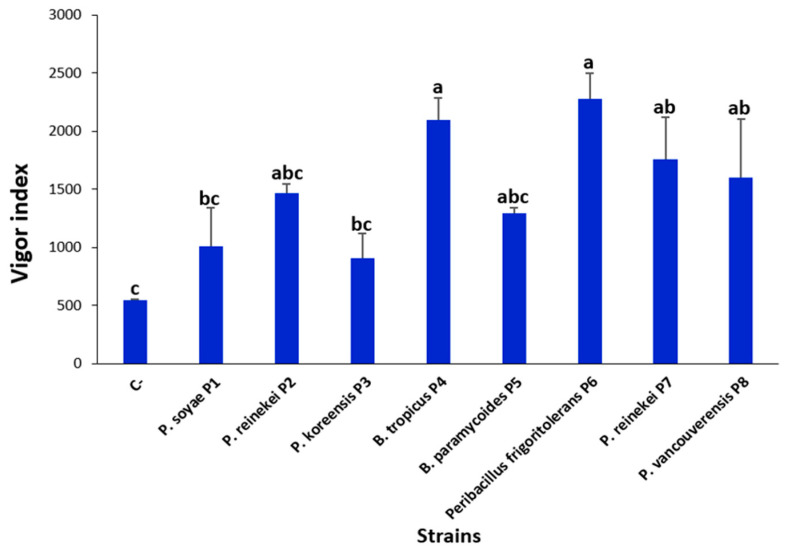
Impact of wheat seed bacterial inoculation on the vigor index. The values indicate means derived from four replicates (*n* = 4). The use of different superscript letters (a, b, c…) signifies the statistically significant differences at a 95% confidence level among the treatments.

**Figure 6 microorganisms-12-01423-f006:**
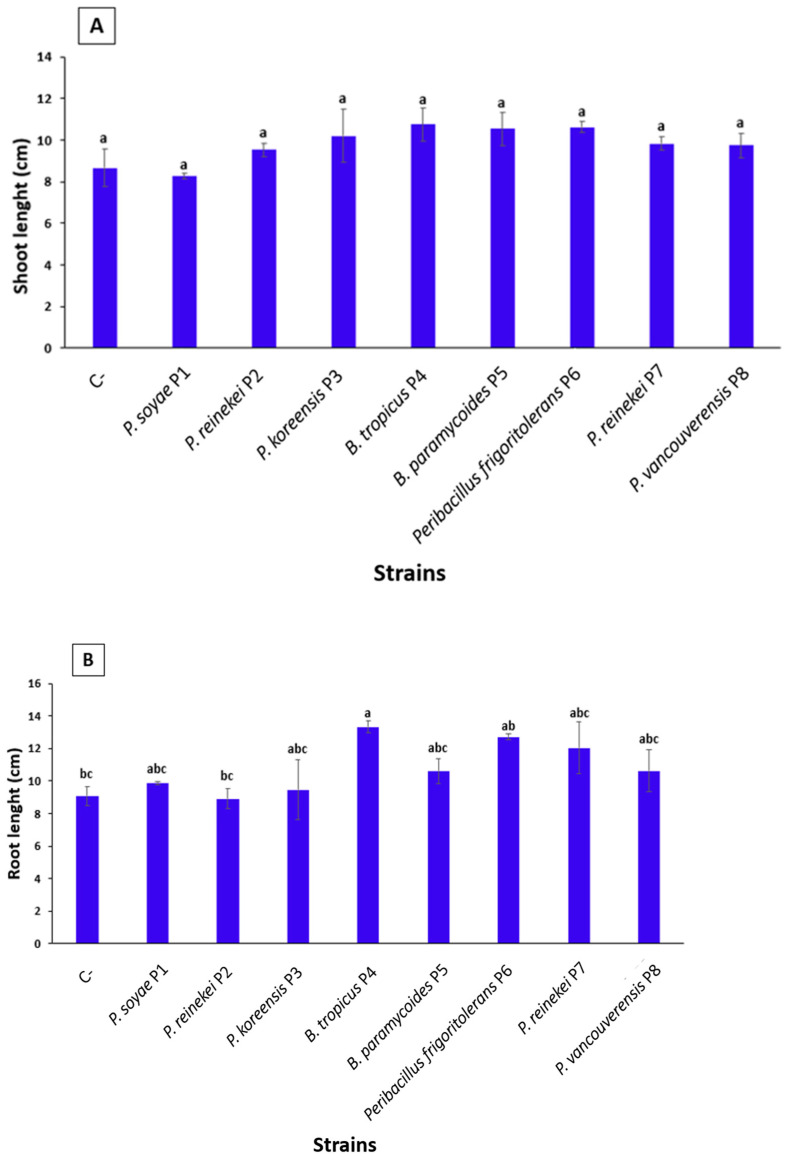
Impact of bacterial strain treatments on shoot (**A**) and root growth (**B**). The values indicate means derived from four replicates (*n* = 3). Using different superscript letters (a, b, c...) signifies the statistically significant differences at a 95% confidence level among the treatments.

**Figure 7 microorganisms-12-01423-f007:**
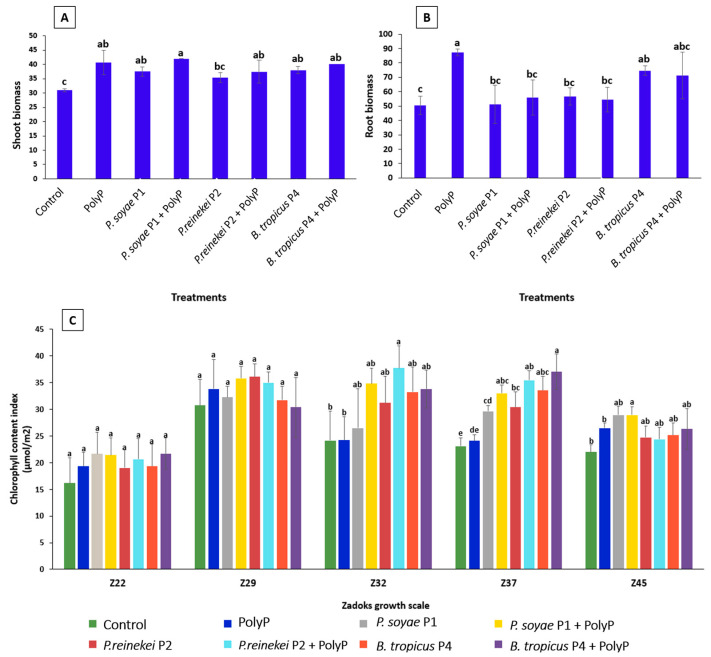
Impact of bacterial inoculation on wheat growth. (**A**) Shoot biomass, (**B**) root biomass at the Z69, and (**C**) chlorophyll content index at Z22, Z29, Z32, Z37, and Z45. The bars display the five replicates’ mean and standard deviation. The different superscript letters (a, b, c, etc.) show statistically significant differences at the 95% confidence level between treatments.

**Table 1 microorganisms-12-01423-t001:** Molecular characterization and additional plant-growth-promoting traits of isolated strains. The level of effectiveness: absent (−), low (+), moderate (++), and strong (+++). Reporter letters in superscript (a, b, c…) indicate a statistically significant difference at the 95% confidence level between treatments.

Strains	Ammonia Production (µmoL/mL)	Zn Solubilization		IAA Production (µg/mL)
		ZnO	ZnCO_3_	
*Pseudomonas soyae* P1	3.3 ± 0.1 ^c^	++	+++	44.1 ± 2 ^a^
*Pseudomonas reinekei* P2	5.1 ± 0.2 ^b^	++	++	9.7 ± 0.1 ^cde^
*Pseudomonas koreensis* P3	1.7 ± 0.1 ^d^	+++	+++	8.6 ± 0.06 ^de^
*Bacillus tropicus* P4	3.01 ± 0.1 ^c^	+	−	7.5 ± 0.3 ^de^
*Bacillus paramycoides* P5	9.1 ± 0.4 ^a^	−	−	9.7 ± 0.07 ^cde^
*Peribacillus frigolitolerans* P6	4.8 ± 0.1 ^b^	−	−v	25.6 ± 10.3 ^b^
*Pseudomonas reinekei* P7	5.6 ± 0.1 ^b^	++	++	17.2 ± 0.8 ^bcd^
*Pseudomonas vancouverenesis* P8	5.6 ± 0.2 ^b^	++	+++	24.8 ± 1.2 ^bc^

**Table 2 microorganisms-12-01423-t002:** Siderophores, HCN production, and bacterial biocontrol activities. The degree of effectiveness can be categorized as follows: absent (−), low (+), moderate (++), and strong (+++). The different superscript letters (a, b, c, etc.) indicate statistically significant differences at the 95% confidence level between treatments.

Strains	Siderophores	HCN	Antifungal (Development Zone (cm))	
			*F. culmorum* Sucardu	*F. culmorum* P.V
Control	−	−	8.5 ± 0.03 ^a^	6.5 ± 0.04 ^a^
*P. soyae* P1	+	+++	0	4.13 ± 0.11 ^d^
*P. reinekei* P2	+++	+++	0	4.53 ± 0.04 ^c^
*P. koreensis* P3	+	+++	4.33 ± 0.11 ^c^	4.54 ± 0.04 ^c^
*B. tropicus* P4	+	+++	3.30 ± 0.13 ^d^	4.00 ± 0.06 ^d^
*B. paramycoide* P5	+	+++	1.66 ± 0.11 ^e^	5.53 ± 0.11 ^b^
*P. frigotolerans* P6	+	++	1.76 ± 0.15 ^e^	4.63 ± 0.08 ^c^
*P. reinekei* P7	++	++	4.33 ± 0.11 ^c^	4.13 ± 0.11 ^d^
*P. vancouverensi* P8	++	+++	5.06 ± 0.08 ^d^	4.3 ± 0.13 ^cd^

**Table 3 microorganisms-12-01423-t003:** Wheat seed germination assay. The various letters in superscript (a, b…) denote the statistically significant difference between treatments at a 95% confidence level.

Strains	Germination Rate (%)
Control	30.76 ^d^
*P. soyae* P1	53.84 ± 10.25 ^bcd^
*P. reinekei* P2	79.48 ± 3.41 ^abc^
*P. koreensis* P3	46.15 ± 10.25 ^cd^
*B. tropicus* P4	87.17 ± 8.54 ^ab^
*B. paramycoides* P5	61.53 ± 5.12 ^abcd^
*P. frigoritolerans* P6	97.43 ± 3.41 ^a^
*P. reinekei* P7	79.92 ± 11.96 ^abc^
*P. vancouverensis* P8	76.92 ± 20.5 ^abc^

## Data Availability

The original contributions presented in the study are included in the article, further inquiries can be directed to the corresponding authors.

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
