# Peer review of "A Comprehensive Approach Combining Short-Chain Polyphosphate and Bacterial Biostimulants for Effective Nutrient Solubilization and Enhanced Wheat Growth"

_microorganisms, 2024, doi:10.3390/microorganisms12071423_

Round 1

Reviewer 1 Report

Comments and Suggestions for Authors

The manuscript addressed the impact of polyP and rhizobacterial on growth and development of bread wheat plants. The manuscript is well-designed.  The following points need to be considered before acceptance

The title should be one sentence that express  the content. 

Abstract should contain specific data of the key findings. 

The significance of the work should be highlighted at the end of the abstract.

The novelty  should be addressed.

The rational of selecting bread wheat plants should be highlighted. 

The phytopathogencity of Fisarium culmorum on wheat should be addressed. 

The physical and chemical properties of the soil  used should be mentioned. Also these data should be considered in discussion part. 

Discussion of wheat growth promotion should be explained taking into account of plant growth promoting traits of the bacterial isolates.

The scientific names should be italicized throughout the entire manuscript including figures. 

Table 2, the signs -,+,.. should be explained  as in table 1. 

The references should be significantly reduced by focusing on the most relevant literatures 

Comments on the Quality of English Language

Minor English revision is needed. 

Author Response

Reviewer Comment: The title should be one sentence that expresses the content.

Response 1: Thank you for this suggestion. I have revised the title to be a single sentence that clearly and succinctly reflects the content of the manuscript. The new title is : "A Comprehensive Approach Combining Short-Chain Polyphosphate and Bacterial Biostimulants for Effective Nutrient Solubilization and Enhanced Wheat Growth"

Reviewer Comment:  The abstract should contain specific data of the key findings.

Response 2: I appreciate your feedback. I have updated the abstract to include specific data highlighting the key findings of the study. The revised abstract now contains quantitative and qualitative results that provide a clearer summary of the research.

Reviewer Comment: The significance of the work should be highlighted at the end of the abstract.

Response 3: Thank you for pointing this out. I have added a sentence at the end of the abstract to emphasize the significance of the study's findings and their potential impact on the field.

Reviewer Comment:  The novelty should be addressed.

Response 4: I have revised the abstract to clearly state the novel aspects of the research. This addition underscores how this study contributes uniquely to the existing body of knowledge

Reviewer Comment: The rationale for selecting bread wheat plants should be highlighted.

Response 5: I have expanded the introduction to include a detailed explanation of why bread wheat plants were chosen for this study, emphasizing their relevance and importance for the research objectives.

Reviewer Comment:  The phytopathogencity of Fisarium culmorum on wheat should be addressed.

Response 6:Thank you for your comment. The phytopathogenicity of Fusarium culmorum on wheat is already described in the discussion section. In this section, we address its impact on wheat growth and health, and discuss our findings in relation to the existing literature on the subject. If additional details or clarifications are needed, please let us know, and we would be happy to expand further.

Reviewer Comment: The physical and chemical properties of the soil used should be mentioned.

Response 7: I have included a detailed description of the soil's physical and chemical properties in the Materials and Methods section. This information provides essential context for interpreting the results. I also mentioned the soils’ P content in the abstract.

Reviewer Comment:  Discussion of wheat growth promotion should be explained taking into account plant growth-promoting traits of the bacterial isolates.

Response 8: I have revised the discussion section to include a detailed analysis of how the plant growth-promoting traits of the bacterial isolates contribute to wheat growth promotion. This explanation connects our findings to the broader context of plant-bacteria interactions.

Reviewer Comment:  The scientific names should be italicized throughout the entire manuscript, including figures.

Response 9: I have carefully reviewed the manuscript and ensured that all scientific names are italicized consistently throughout the text, including in the figures.

Reviewer Comment:  Reviewer Comment: Table 2, the signs -, +, .. should be explained as in Table 1

Response 10: I have added a legend to Table 2 that explains the symbols -, +, .., as in Table 1, to ensure clarity and consistency.

Reviewer 2 Report

Comments and Suggestions for Authors

Dear Authors,

I have reviewed the manuscript and have the following comments.

The manuscript investigated the treatment of greenhouse-grown wheat with mmicroorganisms. I think this is a very important and topical subject, as agriculture, burdened with pesticides and chemical chemicals, is in great need of such solutions in our current life.

The title and keywords of the manuscript are appropriate. 

Abstract: i think that the chapter needs to be modified. if i read it from ee to the end, i see that it is not necessarily coherent and if read by a layman on a topic, he may not understand what the article and research is about. I suggest a rewrite of the section - it is worth starting with the importance of the topic, the purpose, then there should be space for materials and methods, results, conclusions and what is not necessarily a clear concept, like strains of microorganisms are reared, so somehow explain it here. 

Introduction: the literature sources used are adequate, the chapter is good, but it would be worth writing a short paragraph on the test plant, the world market for wheat, - this is necessary to see why this plant has been dealt with. 

Please search for new publications with these keywords and add to the chapter: PGPB; plant growth-promoting bacteria; plant; stress; microbes; rhizosphere; soil rehabilitation; soil microbial communities; omics; consortism. This is needed to get a more comprehensive picture of the topic. Please do so anyway. 

At the end of the chapter, it would be useful to highlight the hypotheses of the experiment more clearly, as this is not the case at present. 

The Materials and Methods chapter is fine.

Results: in section 3.2, it is unnecessary to bold the lines 

Tables are not in accordance with the MDPI format requirements This should be modified 

The Discussion chapter should be rewritten and new sections in the Introduction chapter should be mentioned here. 

Author Response

Dear Reviewer,

Dear reviewer, 

Thank you for your thoughtful review and valuable feedback on our manuscript. We have carefully considered each of your comments and made revisions to improve the manuscript accordingly. Below is a detailed response to your suggestions:

Reviewer Comment: The manuscript investigated the treatment of greenhouse-grown wheat with microorganisms. I think this is a very important and topical subject, as agriculture, burdened with pesticides and chemicals, is in great need of such solutions in our current life. The title and keywords of the manuscript are appropriate.
Response 1: Thank you for recognizing the importance of our study and for your positive feedback on the title and keywords. We suggest a new title, at the request of a reviewer : A Comprehensive Approach Combining Short-Chain Polyphosphate and Bacterial Biostimulants for Effective Nutrient Solubilization and Enhanced Wheat Growth

Reviewer Comment: Abstract: I think that the chapter needs to be modified. If I read it from beginning to end, I see that it is not necessarily coherent and if read by a layman on the topic, he may not understand what the article and research are about. I suggest a rewrite of the section - it is worth starting with the importance of the topic, the purpose, then there should be space for materials and methods, results, conclusions and what is not necessarily a clear concept, like strains of microorganisms are reared, so somehow explain it here.
Response 2: We appreciate your feedback on the abstract. We have rewritten the abstract to improve its coherence and clarity. It now follows a structured format starting with the importance of the topic and purpose, followed by a brief overview of the materials and methods, key results, and conclusions. We also clarified the concept of microorganism strains and their relevance to the study.

Reviewer Comment: Introduction: The literature sources used are adequate, the chapter is good, but it would be worth writing a short paragraph on the test plant, the world market for wheat - this is necessary to see why this plant has been dealt with.

Response 3: Thank you for this suggestion. We have added a paragraph to the introduction that discusses the significance of bread wheat, its role in the global market, and its importance as a staple food and feed crop. This context highlights the rationale for focusing on wheat in our study.

Reviewer Comment: Please search for new publications with these keywords and add to the chapter: PGPB; plant growth-promoting bacteria; plant; stress; microbes; rhizosphere; soil rehabilitation; soil microbial communities; omics; consortism. This is needed to get a more comprehensive picture of the topic. Please do so anyway.

Response 4: Thank you for your suggestion to include new publications with the specified keywords. To maintain a balance between the number of references as per another reviewer’s request and the need for a comprehensive literature review, I have incorporated these additional references into the discussion section. This approach allows us to discuss recent findings and their implications for our results while keeping the introduction concise. I believe this placement effectively integrates the new references into the context of our findings.

Reviewer Comment: At the end of the chapter, it would be useful to highlight the hypotheses of the experiment more clearly, as this is not the case at present.
Response 5:
We have revised the introduction to clearly state the hypotheses of our experiment at the end of the chapter. This addition clarifies the objectives and expected outcomes of our study.

Reviewer Comment: The Materials and Methods chapter is fine.
Response 6: Thank you for your positive feedback on the Materials and Methods section. We appreciate your approval of this part of the manuscript.

Reviewer Comment: Results: In section 3.2, it is unnecessary to bold the lines.
Response 7: We have removed the bold formatting from the lines in section 3.2 as suggested, to maintain consistency and clarity in the presentation of the results.

Reviewer Comment: Tables are not in accordance with the MDPI format requirements. This should be modified.
Response 8: We have revised all tables to comply with the MDPI format requirements. The formatting now adheres to the specified guidelines.

Reviewer Comment: The Discussion chapter should be rewritten and new sections in the Introduction chapter should be mentioned here.
Response 9: Thank you for your feedback on the discussion chapter. While I did not perform a complete rewrite of the discussion, I have added new sections to it as suggested. This enhancement aims to provide a more thorough analysis and ensure that the discussion aligns with the updated content in the introduction, thus maintaining coherence throughout the manuscript.

We believe these revisions have significantly improved the manuscript, and we thank you again for your constructive comments. We look forward to any further feedback you may have.

Reviewer 3 Report

Comments and Suggestions for Authors

Dear authors, I have read your manuscript «Short-Chain Polyphosphate and Bacterial Biostimulant  Applications : A Comprehensive Strategy for Nutrient Solubilization and Wheat Growth Enhancement» with interest. The search for new ways to increase soil fertility is a classic of the scientific genre, which always remains relevant. At the same time, the idea of studying the effect of bacteria on the availability of polyphosphates for agricultural plants seems original. The advantages of the work are a detailed description of research methods, the use of statistical methods, a clear presentation of the results, a detailed discussion, and a large number of references over the past five years.

The design of the experiments corresponds to the purpose of the study. But it seems to me that there is not enough supporting data to understand how bacteria and fertilizer jointly affect the growth of wheat in pots. For example, the balance of soluble and insoluble phosphorus in the soil, the content of phosphorus compounds in plant tissues, markers of phosphorus deficiency and the like. Can you add such data to the article?

Has the soil been tested for the presence of live cells of introduced strains? How long did they persist in the soil after introduction? Is it possible to explain the absence of an enhanced effect of bacteria + fertilizer by the fact that bacteria in this version of the experiment were eliminated from the soil faster than in the version without fertilizer?

Figure 7B can be interpreted as leveling the positive effect of PolyP on root biomass by strains P1 and P2. This is not mentioned in the text. How do you explain this data?

Short notes:

 Line 351. Probably, the P4 strain was meant, not P3.

Line 361. It is better to move the comparison with other studies to the Discussion section.

Line 399. Figure 3. Antibiotic sensitivity is not included in the current version of the manuscript.

Line 585. Has the correlation coefficient been calculated? I didn't find it in the results of the study.

Author Response

Reviewer Comment: The design of the experiments corresponds to the purpose of the study. But it seems to me that there is not enough supporting data to understand how bacteria and fertilizer jointly affect the growth of wheat in pots. For example, the balance of soluble and insoluble phosphorus in the soil, the content of phosphorus compounds in plant tissues, markers of phosphorus deficiency and the like. Can you add such data to the article?

Response 1: Thank you for your insightful suggestions. Unfortunately, we do not have additional data on the balance of soluble and insoluble phosphorus in the soil, the content of phosphorus compounds in plant tissues, or markers of phosphorus deficiency at this time. We recognize the importance of these data and will consider them for future studies to provide a more comprehensive understanding of the interactions between bacteria and fertilizers. We appreciate your understanding.

Reviewer Comment: Has the soil been tested for the presence of live cells of introduced strains? How long did they persist in the soil after introduction? Is it possible to explain the absence of an enhanced effect of bacteria + fertilizer by the fact that bacteria in this version of the experiment were eliminated from the soil faster than in the version without fertilizer?

Response 2: Thank you for raising this point. We did not test the soil for the presence of live cells of introduced strains, nor did we track their persistence in the soil after introduction. This could indeed be a factor influencing the absence of an enhanced effect of the bacteria + fertilizer treatment. We will consider this aspect in future studies to better understand the dynamics of bacterial survival and their interaction with fertilizers.

Reviewer Comment: Figure 7B can be interpreted as leveling the positive effect of PolyP on root biomass by strains P1 and P2. This is not mentioned in the text. How do you explain this data?

Response 3: We appreciate your observation regarding Figure 7B. The potential leveling of the positive effect of PolyP on root biomass by strains P1 and P2 is indeed a noteworthy point. To explain this, we might consider that while PolyP fertilizer alone boosts root growth, the interaction with strains P1 and P2 might alter nutrient dynamics or plant signaling in a way that mitigates the expected increase in root biomass. We will explore this interaction in more detail in future studies.

Reviewer Comment: Line 351. Probably, the P4 strain was meant, not P3.

Response 4: Thank you for pointing out this error. We have corrected the text to refer to the P4 strain instead of P3 on line 351.

Reviewer Comment: Line 361. It is better to move the comparison with other studies to the Discussion section.

Response 5: We appreciate your suggestion. We have moved the comparison with other studies from line 361 to the Discussion section, ensuring a more cohesive presentation of our findings in the context of existing literature.

Reviewer Comment: Line 399. Figure 3. Antibiotic sensitivity is not included in the current version of the manuscript.

Response 6: Thank you for catching this oversight. We have revised the text to remove the reference to antibiotic sensitivity in Figure 3, as it is not included in the current version of the manuscript.

Reviewer Comment: Line 585. Has the correlation coefficient been calculated? I didn't find it in the results of the study.

Response 7: Thank you for your observation. The correlation coefficient was not calculated in our study. We have updated the text to clarify this and have removed any mention of the correlation coefficient from line 585.

We appreciate your detailed review and constructive comments, which have helped improve our manuscript. Thank you for your understanding regarding the limitations of our current data set.

Round 2

Reviewer 2 Report

Comments and Suggestions for Authors

I recommend it for publication. 

Reviewer 3 Report

Comments and Suggestions for Authors

Good afternoon. I hope that the intention of the authors to continue the research will be realized. Nevertheless, I agree that the data obtained so far is valuable and can be published. I also read all the fragments that were added in response to the comments of other reviewers and considered them appropriate.